# High-resolution structural and functional retinal imaging in the awake behaving mouse

Guanping Feng[1,2], Aby Joseph[2,3], Kosha Dholakia[1,2], Fei Shang[2,4], Charles W. Pfeifer [2,5], Derek Power[2], Krishnan Padmanabhan[2,4,6] & Jesse Schallek [2,4,5✉]

The laboratory mouse has provided tremendous insight to the underpinnings of mammalian central nervous system physiology. In recent years, it has become possible to image single neurons, glia and vascular cells in vivo by using head-fixed preparations combined with cranial windows to study local networks of activity in the living brain. Such approaches have also succeeded without the use of general anesthesia providing insights to the natural behaviors of the central nervous system. However, the same has not yet been developed for the eye, which is constantly in motion. Here we characterize a novel head-fixed preparation that enables high-resolution adaptive optics retinal imaging at the single-cell level in awake-behaving mice. We reveal three new functional attributes of the normal eye that are overlooked by anesthesia: 1) High-frequency, low-amplitude eye motion of the mouse that is only present in the awake state 2) Single-cell blood flow in the mouse retina is reduced under anesthesia and 3) Mouse retinae thicken in response to ketamine/xylazine anesthesia. Here we show key benefits of the awake-behaving preparation that enables study of retinal physiology without anesthesia to study the normal retinal physiology in the mouse.

[1] Department of Biomedical Engineering, University of Rochester, Rochester, NY 14620, USA. [2] Center for Visual Science, University of Rochester, Rochester, NY 14627, USA. [3] The Institute of Optics, University of Rochester, Rochester, NY 14620, USA. [4] Department of Neuroscience, University of Rochester, Rochester, NY 14642, USA. [5] Flaum Eye Institute, University of Rochester, Rochester, NY 14642, USA. [6] Intellectual and Developmental Disabilities Research Center, University of Rochester, Rochester, NY 14642, USA. ✉email: jschall3@ur.rochester.edu

The laboratory mouse is an indispensable model for biomedical research due to its size, accessibility, sequenced genetic catalog, and ability to model aspects of human disease. In particular, it has enabled the study of the anatomy and function of the mammalian eye, which, other than size and notable lack of fovea, resembles in many ways the human eye[1]. To achieve high-resolution retinal imaging in the mouse, anesthesia is usually required to stabilize the preparation and suppress eye motion that makes cellular-level functional assessments nearly impossible[2]. Some approaches have demonstrated that mouse retinal imaging is possible with hand-restraint[3,4], however, the utility of this approach is for single-snapshot photographic purposes and do not provide a stable optical axis that is essential for functional measures.

Delivering general anesthesia provides a stabilized in vivo preparation and mitigates eye motion; however, it may also alter normal physiologic function, thereby limiting the interpretation of in vivo measurements, especially those in the central nervous system (CNS) function[5,6]. To this end, behavioral and physiological neuroscientists have developed head-fixed preparations to stabilize the brain for electrophysiology and in vivo microscopy[7], obviating the need for anesthesia. Notably, studies have reported various key neurophysiological differences in the awake versus the anesthetized state[8,9]. Another consequence of anesthesia for vision research is that it removes the natural ocular movement that provides spatiotemporal contrast to the visual system. Suppression of natural eye motion fundamentally changes the spatiotemporal kinetics of ganglion cell output to the lateral geniculate nucleus, superior colliculus, and visual cortex studies in mice[10]. There are also reports that locomotion in the awake preparation substantially changes the physiological response of the visual cortex[11], yet mechanisms are not fully understood. Therefore, leaving eye motion intact could further advance the understanding of mouse oculomotor behavior and, in particular, how biological eye motion may impact and drive basic visual physiology.

In addition to the benefits of preserving eye motion and eliminating the confounds of anesthesia, imaging the awake mouse may also aid retinal imaging in several additional aspects. First, imaging the awake animal can prevent optical opacification from prolonged anesthesia, which has been a tremendous challenge for ocular imaging in the anesthetized mouse[12]. Secondly, thermoregulation is not necessary when imaging the awake mouse, which has shown to impact homeostatic physiology[13]. And finally, the awake mouse maintains normal eye clarity by blinking and constantly refreshing the tear film without the need for contact lens or lubrication, which could confound behavioral or natural optical conditions[14].

Here we recognize many potential benefits of studying the awake mouse retina at high resolution without anesthesia and overcome a number of the above limitations by developing and characterizing a head-restrained preparation for retinal imaging. We provide an open-source, fully 3D-printed apparatus to hold the mouse head to provide a stable pupil with a body suspended above a rotational cylinder that allows free ambulation and simultaneous measurement of ambulatory velocity and behavior. We measure the stability of the optical axis that facilitates high-resolution structural and functional retinal imaging. We further show the utility by imaging the mouse eye with a leading commercial multimodal scanning laser ophthalmoscopy (SLO) and optical coherence tomography (OCT). We then demonstrate the benefits of high-resolution imaging using a custom adaptive optics scanning light ophthalmoscope (AOSLO) to visualize various retinal cellular structures at micron-level resolution. The stability of the eye and image quality in the awake mouse are validated for both SLO and AOSLO imaging. Using high-speed line-scanning AOSLO high-speed line-scanning AOSLO imaging[15,16], we discover a previously overlooked micro-tremor-like eye motion. In this early work, we showcase several scientific opportunities by highlighting three key physiologic changes in the retina induced by anesthesia.

## Results

**Mice acclimate to head-restrained apparatus within ~1 week**. The general approach and setup is presented in Fig. 1a. The mouse cranium was held stable through an affixed headplate. The Y-shaped headplate was implanted in the center of the mouse cranium in parallel to the anterior-posterior axis after a single surgical cut through the scalp under anesthesia (Fig. 1b). All mice survived the procedure. Headplated mice exhibited normal behavior and maintained a healthy outward appearance with groomed hair coat and normal activity level. No inflammation or other side effects related to the headplate implantation were observed. The weights of all mice also remained consistent pre- and post-surgery. Within 3 days of the surgical placement of the headplate, head-restrained mice became acclimated to the apparatus over five training sessions lasting 30–60 min. The Y-shaped headplate was mounted to a fixed arm suspended over a rotational cylinder such that the mice were able to freely ambulate fore-and-aft. After training and acclimation in the head-fixed apparatus, mice did not exhibit aversion or retreat response maintained normal grooming behavior (Supplementary Video 1), further indicating a calm baseline state with no outward signs of distress[17,18]. The headplate surgery left the eye open with a normal blink reflex (Supplementary Video 1).

**Retinal imaging with commercial multimodal SLO + OCT in awake, behaving mice**. The head-restrained apparatus enabled high-quality structural and functional retinal imaging using commercial SLO and OCT imaging platforms without the aid of adaptive optics (Fig. 1d, e and Supplementary Video 3). All leading fluorescent, autofluorescence, and reflectance imaging modalities were possible using the commercial SLO + OCT system (Heidelberg Spectralis, Heidelberg Engineering, Germany). Using reflectance, major superficial retinal blood vessels and nerve fiber bundles were revealed under both NIR and blue channels. Popular angiography contrast agents could also be used in the awake mouse to reveal sodium fluorescein angiography (FA) with clear arteriole, venule, and capillary imaging of the retinal circulation using 488 nm excitation as well as ~500 nm long-pass fluorescence. Indocyanine green angiography (ICGA) using NIR light revealed both the superficial retinal circulation as well as demonstrated good penetrance to visualize the choroidal circulation. Visualizing the choroid is notably challenging in the C57BL6/J mouse as the RPE is densely pigmented. To show the potential for imaging transgenically labeled fluorescence, we also imaged fluorescent microglia using CX3CR1 mice with an enhanced green fluorescent protein (EGFP) in cells that contain the chemokine CX3 receptor 1. In all SLO imaging modalities (fluorescence and reflectance), the single-frame image contrast and signal-to-noise ratio (SNR) was sufficient to perform registration at a frame rate of up to 8.8 Hz with the built-in auto-registration software (Supplementary Video 2). The en face registration enabled real-time B-scan registration for volumetric OCT cubes and improved SNR by averaging multiple images (ART, automatic real-time tracking)[19]. We imaged a 3D OCT cube with 96 B-scans. Using built-in real-time registration and ART, the OCT cube revealed clear delineation of retinal layers comparable to that achieved with anesthesia despite residual gaze shifts and eye motion within the acquisition of the full cube.

**Head-restrained mouse provides a stable optical axis for high-resolution retinal imaging**. A major challenge for high-

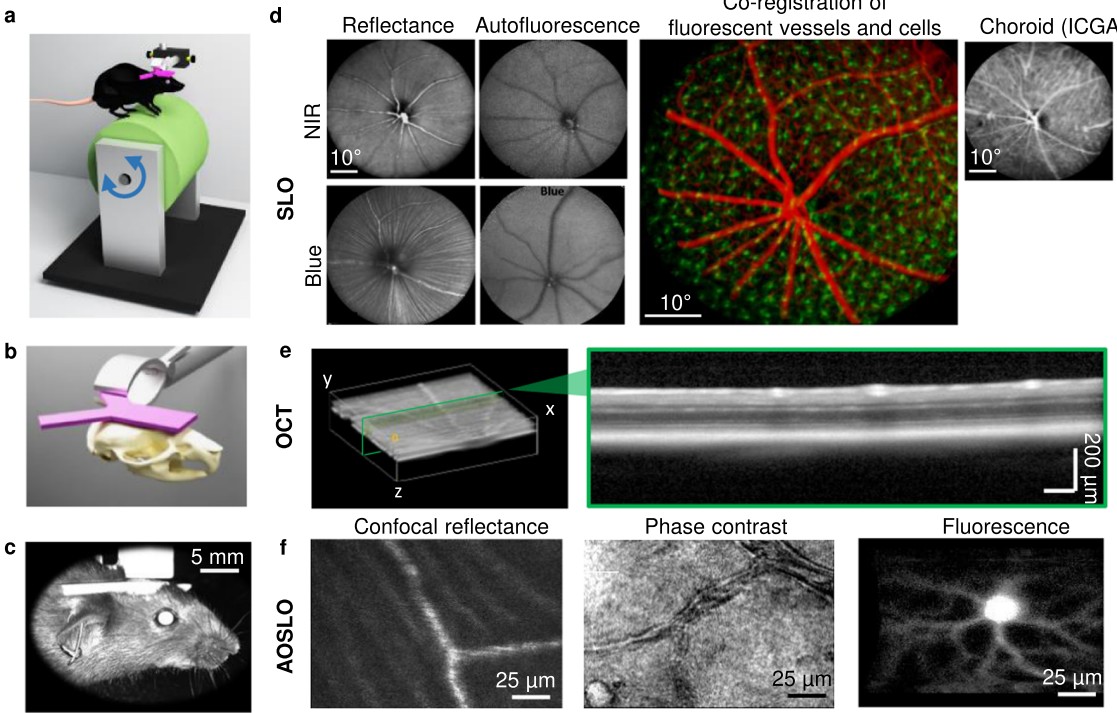

**Fig. 1 Structural and functional retinal imaging for the awake, behaving mouse using head-restrained preparation. a** Head-restrained preparation for awake, behaving mouse retinal imaging. A surgical implanted Y-shape headplate on the mouse cranium restrains head motion, while ambulation is allowed on a rotational cylinder during imaging. **b** Schematic showing the position of the headplate on the center of the mouse cranium in parallel with the anterior-posterior axis. **c** A photograph of the head-restrained mouse imaged with commercial SLO. **d** Multimodal cSLO image of the retina in the head-restrained awake mice. Different retinal structures are visualized (Left - right): reflectance and autofluorescent imaging reveal major superficial retinal blood vessels and nerve fiber bundles under both NIR (top) and blue (bottom) channels. Fluorescent co-registration of NIR excited indocyanine green angiography (ICGA) of superficial blood vessel networks (rendered as red pseudo color) with blue excited fluorescence-labeled immune cells (rendered as green pseudo color) in a transgenic CX3CR1 mouse. ICGA of deep choroidal blood vessel network. **e** OCT cube and a representative b-scanned cross-section imaged in a head-restrained awake mouse. **f** AOSLO images show multimodal capabilities. Left: confocal reflectance image reveals nerve fiber bundle and retinal capillaries. Middle: phase-contrast imaging of a microcyst and a branch of retinal blood vessels. Right: fluorescence image of a YFP-labeled retinal ganglion cell with dendrites and axons highlighted.

resolution imaging of the awake mouse retina is that the imaging beam entering the pupil of the mouse eye (~2 mm) is sensitive to head and eye motion. Pupil misalignments can cause clipping and vignetting that seriously impact wavefront sensing, blur the image, and create unstable light delivery/collection for functional measurements. For most high-resolution applications, a fully dilated pupil is desirable because it maximizes the numerical aperture (NA ~0.49), thereby improving imaging resolution and light collection efficiency. However, it also creates a requirement of stability to align the optical pupil of the instrument to the pupil of the animal. To quantify the optical axis stability for retinal imaging, the exit pupil of the eye in five awake mice was imaged with SLO for over 20 min. The pupil was tracked and segmented from the SLO images using custom-built software (Fig. 2a and Supplementary Video 3). The mouse pupil was tracked to quantify beam spatial clipping based on a simulated 2.0 and 1.6 mm diameter imaging beam (Fig. 2b, c). With the 2.0 mm beam, four mice showed excellent pupil stability with <1% averaged area of the pupil being clipped over time. One animal showed more beam clipping (6.65%). We found this to be an outlier due to a suboptimal headplate surgery which left for incomplete eye opening due to minor eyelid droop. Using the smaller 1.6 mm imaging beam, all mice showed negligible beam clipping, representing $0.112 \pm 0.18\%$ of the area of the pupil (Mean ± SD, $n = 5$ mice). To evaluate the temporal impact of beam clipping (the fraction of time that the beam was clipped by >10%), we evaluated SLO videos captured at 8.8 frames

per second (fps). We found $94.75 \pm 11.13\%$ of the frames were unclipped for the 2.0 mm beam (Mean ± SD, $N = 5$ mice) and $99.78 \pm 0.30\%$ of frames were unclipped when using a 1.6 mm beam. The small fraction of pupil clipping in either the spatial or temporal analysis was attributed mostly to gaze behavior of the mouse rather than lack of stability of the headplate preparation. The pupil stability was also examined by comparing the pupil position to relative to the simultaneously recorded gait velocity. There was no correlation between the beam clipping or pupil centration with the locomotion behavior. This suggests pupil stability was attributed to a stably fixed headplate (Fig. 2d). Both the spatial and temporal analysis suggest that the pupil is stable, even under locomotion up to 0.8 m/s (approximately ¼th the top speed of an unrestrained mouse). Thus, the awake mouse eye preparation lends itself favorable for continuous retinal imaging that facilitates functional optophysiology in more natural conditions.

**Gaze shift and eye blinks measured with SLO**. In addition to the movement of the pupil, ocular motion, including gaze shifts and blinks, can disrupt stable imaging approaches. We, therefore, quantified gaze shift and blinking by imaging the retina with SLO for 20 min in five awake mice. Compared to humans, blinking behavior was far less frequent in mice ($1.78 \pm 0.8$ blinks/minute, $n = 5$ mice) compared to that of humans (~20 blinks/min)[20]. The measured duration of the mouse eye blinking was <340.9 ms which corresponds to approximately three frames in the SLO

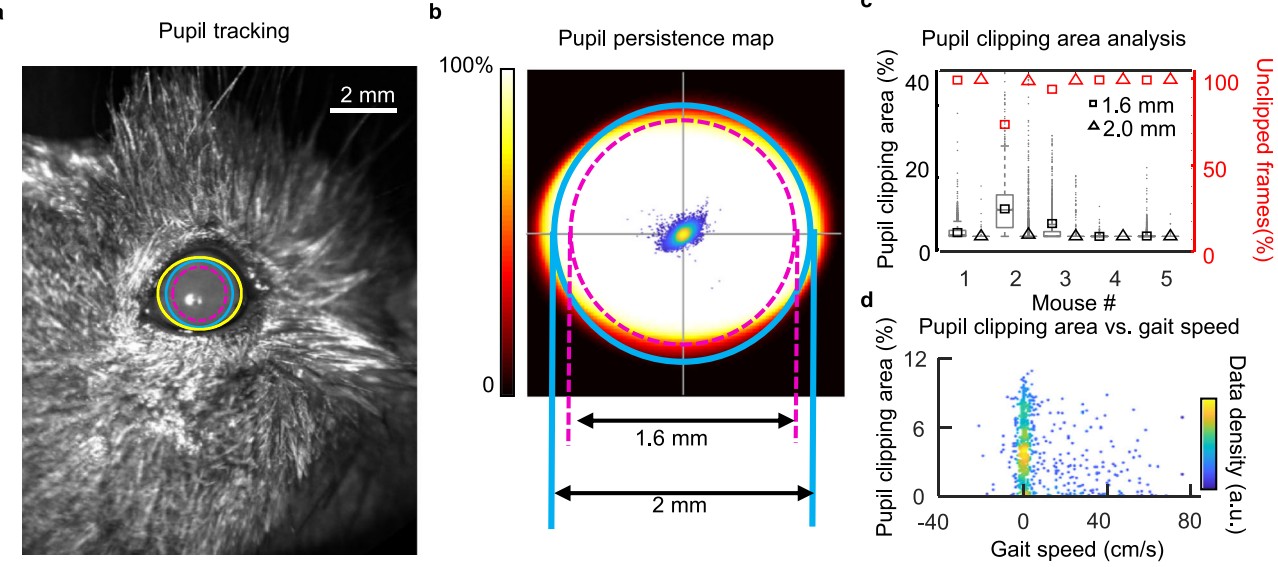

**Fig. 2 Evaluation of pupil stability of head-restrained awake, behaving mice. a** Image of a mouse pupil captured with commercial SLO. The yellow solid circle indicates the tracked pupil boundary of the eye. Cyan solid and dashed circles, respectively indicate the simulated 2 and 1.6 mm diameter imaging beacon of a high-resolution imaging system. **b** Mapping of pupil persistence reveals a stable pupil over 20 min when using 2 mm (solid) and 1.6 mm (dashed) diameter system pupils. **c** The left y-axis shows the pupil clipping in the area and the right y-axis shows the faction of unclipped (<10% area) frames over the 20-minute imaging session (N = 8956 frames). **d** No obvious correlation was observed when correlating the mouse eye motion and gait speeds, which shows consistent optical stability no matter the ambulating state of the mouse.

system. The rare blinking events in mouse facilitate uninterrupted imaging beyond that which is possible in human retinal imaging. Gaze shifts were quantified by tracking retinal features over a 55° FOV (8.8 fps). Mice freely viewed the visual scene of the experimental setup with infrequent, but directed eye motion. This was measured free of head orientation as the head-fixed apparatus constrained gaze to only eye motion. The majority of retinal gaze shifts were in the horizontal axis spanning an observed peak-to-peak range of ±18.00° FOV in free-viewing conditions. This was large compared to ±3.44° of tracked motion in the vertical dimension (Fig. 3a and Supplementary Video 3). Similar to previous studies[21], we found the predominant gaze shifts were >5°, rapid (~50°/s), and rare (~7 per minute in mouse, compared to multiple saccades per second in the human). As mice lack a fovea, these gaze shifts are not true saccades that re-center the image on the fovea[22], but may instead represent a redistribution of the visual scene on areas denser with photoreceptors or smaller ganglion cell receptive fields which reside near the optical axis of the eye[23]. Twenty minutes of semi-continuous video tracking of the mouse retina, gaze behavior showed a clustered pattern of persistence over several preferred gaze directions suggesting a natural resting position of the eye, or preferred gaze direction based on visual features within the laboratory room. To determine the persistence of the gaze positions, the data was then split and normalized to the local mean position for every 10 s. Using this analysis, we found the retinal position stayed within 5° of the visual angle 80.02 ± 0.065% of the time within the 10-s windows, which corresponds to the typical video acquisition window of high-resolution AOSLO imaging. This is relevant for high-resolution imaging as the subtended field for AOSLO imaging is typically 5°, suggesting that offline image registration may correct motion by strip or frame registration approaches without "frame-out" errors which make image registration based on common features or cross-correlation approaches challenging[24].

Locomotion data was also simultaneously recorded using the rotary encoder to investigate the potential correlation between locomotion and gaze behavior. Gait speed and eye movement have weak negative correlations, with the exception of mouse #4,

which had a weak positive correlation. (R = 0.0226; 0.0030; 0.1294; 0.0060; 0.0135, respectively). We observed no obvious change of gaze behavior while animals were at rest or in motion (Fig. 3c). This suggests not only stable image recordings independent of the locomotion of mice, but also a lack of enhanced or suppressed eye motion in correspondence with gait.

**AO corrected wavefront in the awake-behaving mouse is stable over hours.** The paucity of blinks combined with the stability of the optical axis and pupil aperture in the awake-behaving mouse eye made it feasible to perform wavefront sensing and wavefront correction using closed-loop adaptive optics. Using the 2.0 mm entrance pupil of the dilated mouse eye, we found the Hartman–Shack (HS) wavefront sensing spots remained bright and with minimal scatter enabling correction for over 75 min (Fig. 4a–d). This preparation was comparable to the optimal anesthetized preparation[14,25]. Stable HS spot and pupil position enabled active adaptive optics correction with negligible pupil clipping. Wavefront correction voltages applied to the deformable mirror were within the dynamic range of the deformable mirror, indicating sufficient stroke. With closed-loop AO correction, retinal image contrast and signal-to-noise ratio (SNR) remained relatively stable over recordings lasting over 1 h without noticeable degradation of image quality over time (Fig. 4e).

**Cellular resolution AOSLO imaging in the awake mouse.** After AO correction, high-resolution AOSLO imaging revealed cellular retinal structures in the awake mouse, mirroring what was possible in previous studies but without anesthesia (Fig. 1f). Using 796 nm NIR confocal imaging, structures such as single nerve fiber bundles and capillaries less than 5 µm[14] were visible. Using NIR phase-contrast imaging, translucent structures such as single blood cells[16] and vessel walls[16] were visualized. This enabled label-free blood flow measurement (described below). To show fluorescence capabilities, we imaged the Thy-1 YFP-labeled ganglion cells and their dendritic arbors[14] under an excitation of 488 nm. The AOSLO images from all three imaging modalities

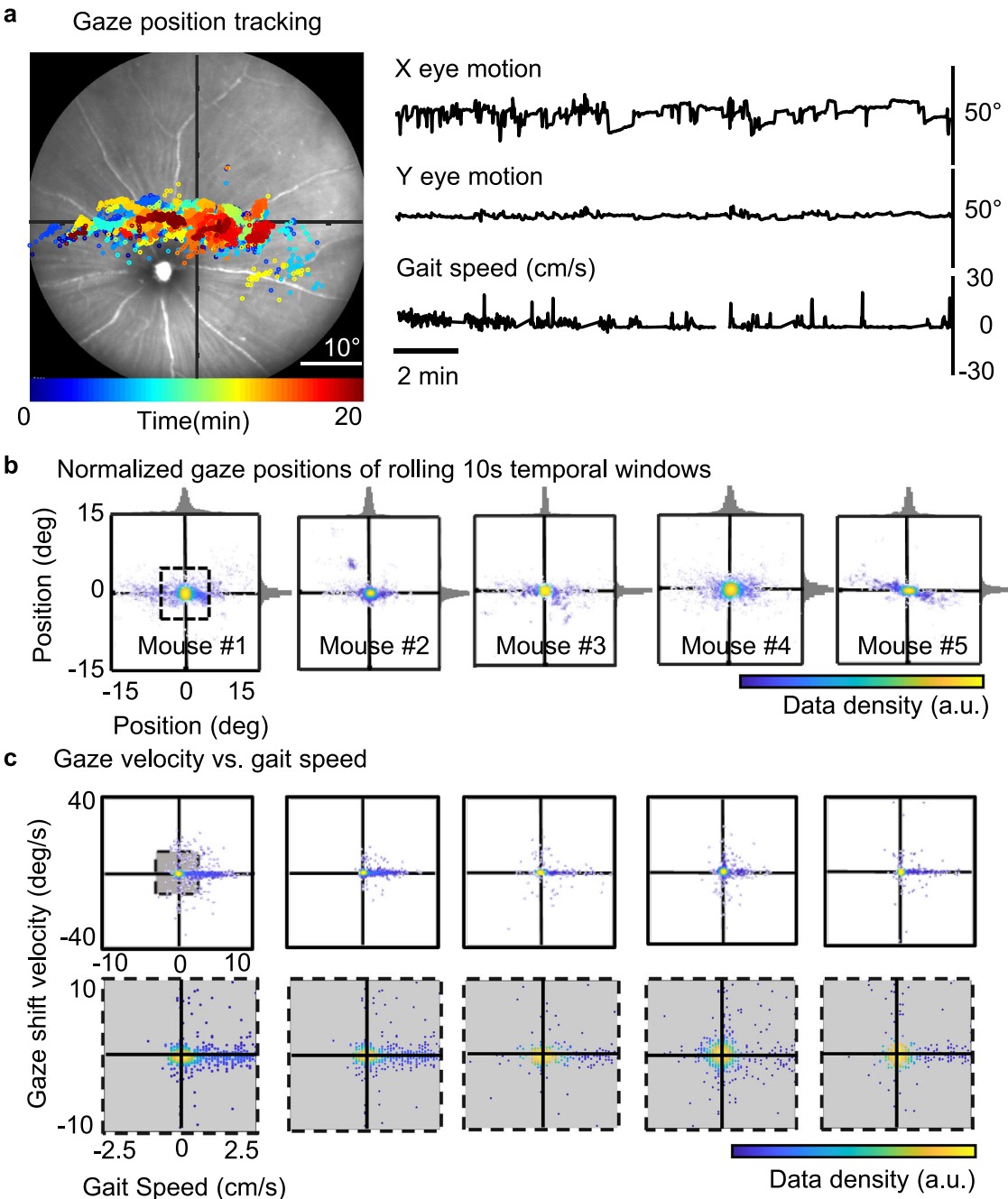

**Fig. 3 Evaluation of gaze behavior in the head-restrained mice. a** Mouse gaze position is measured by tracking retinal offsets. Right: tracked gaze position of a head-restrained mouse over 20 min overlaid on a representative frame of the retina. Left: The 20-min motion trace shows gaze shifts in horizontal (top) and vertical (middle) directions. Simultaneously recorded gait speed with a rotary encoder on the wheel (bottom). **b** Gaze positions of all five mice normalized for every 10 s sliding window over 20 min. Histograms of gaze position distribution overlaid on the horizontal and the vertical axes show a high probability of retinal offsets <5° (labeled as a dashed box) within the 10-second imaging window. **c** Correlations of the mouse gaze velocity and the gait speed, bottom panel shows the zoom-in of the completed dataset labeled as a gray box on the top for visualization.

show contrast and resolution comparable to those previously captured under anesthetized mice. All the NIR phase-contrast, confocal reflectance, and fluorescence imaging modalities were conducted using safe levels of light power without observation of behavioral distress to the imaging light (200–500 µW at the cornea for NIR, 220–330 µW at the cornea for fluorescence). While NIR wavelengths are not expected to be visible to the mice, even the use of bright visible wavelengths did not induce wincing or excessive blinking behavior in the mouse, suggesting these light levels do not induce overt stress or photophobia. Total eye motion in preferred gaze directions were generally held within 5°

of retinal motion, meaning that 'frame-out' errors in image registration were infrequent. Remaining high-frequency, low-magnitude corrections were observed and corrected (Supplementary Video 5).

**Retinal position measured with micron-level precision with the aid of adaptive optics and motion correction.** Using AOSLO imaging, we found the retinal motion was characterized by gaze shifts, slow drifts, and a previously unreported high-frequency, low-amplitude eye motion in the awake mouse. Visual inspection

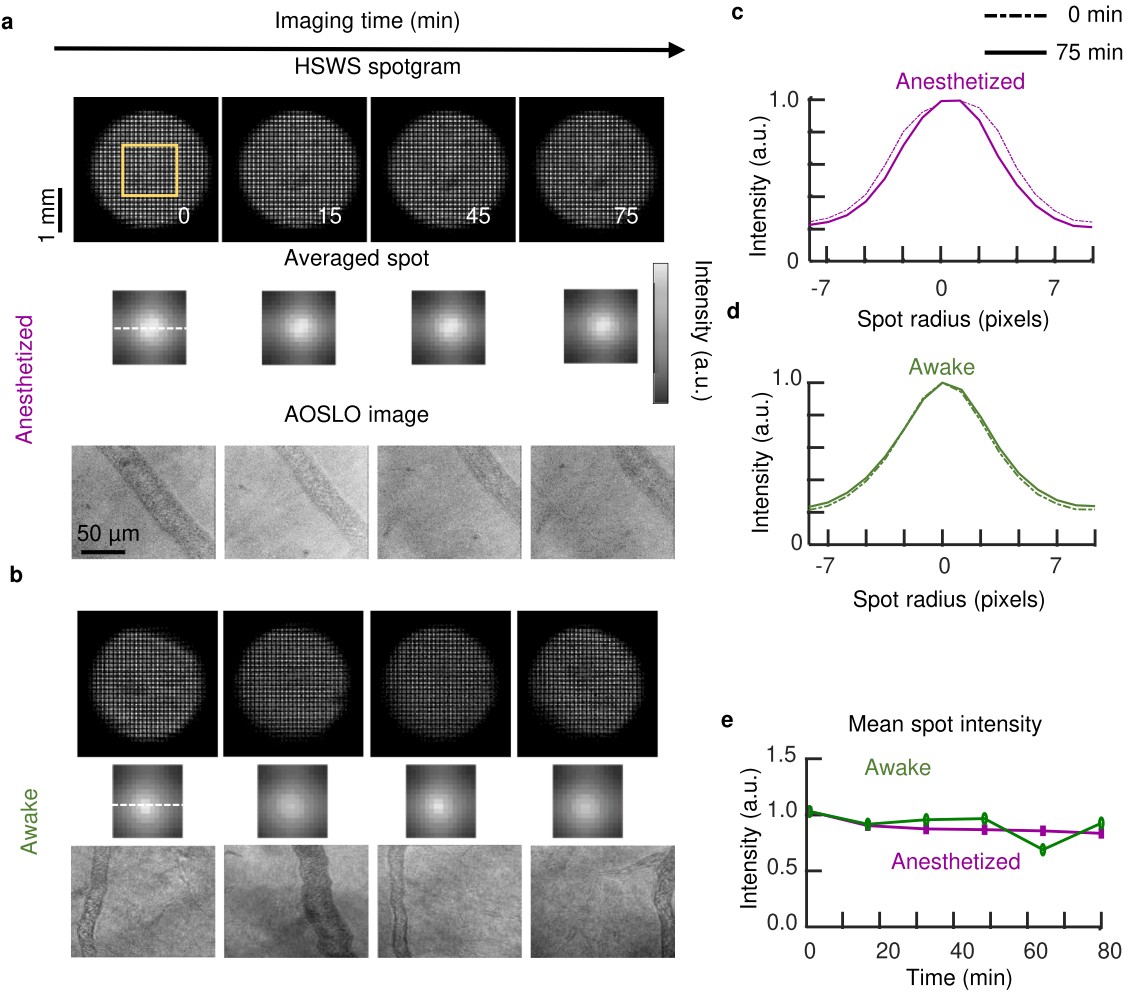

**Fig. 4 Evaluation of AO wavefront correction in the head-restrained awake, behaving mice. a, b** AO corrected imaging in the anesthetized (**a**) and awake (**b**) mice over 75 min. Top: Pupil wavefront spotgram captured with HSWS. Middle: mean spot intensity profile of region labeled with the orange rectangle). Bottom: AOSLO image. The intensity of the mean spot patterns was respectively normalized to the peak intensity at the 0 time points in the awake and anesthesia data. **c, d** Comparing the intensity profiles of the averaged spots (labeled as dashed lines in **a, b**) at 0 min and 75 min. All intensity profiles were respectively normalized to their peak values. In the anesthetized mouse, the spot was slightly widened in 75 min, while the awake mouse was relatively stable. **e** Plot of mean peak intensity over time. The awake mouse showed relatively stable mean peak intensity as well as the anesthetized mice with optimal preparation.

of video-rate data reveals that single frames exhibit fast wobble/shear within single frames (~40 ms, described further below) and larger eye gaze motions that translate the field. Left uncorrected, such eye motion will impart motion distortion/blur, even if imaging with diffraction-limited optics (Supplementary Video 5). A strip-based registration parallel to the fast scan axis (15 kHz)[26] was deployed to measure and correct for the line-by-line motion observed in the awake mouse AOSLO imaging (Fig. 5a and Supplementary Video 6). To show the benefit of the strip/line-based registration approach, we show a post-processing digital correction of the rapid eye motion with greatly improved contrast and sharpness of fine-detailed features over the raw data and rigid-body frame registration approach (Fig. 5b). All fluorescence images presented here have sufficient contrast to perform strip registration in each high-resolution imaging modality without requiring a dual-registration strategy[24]. We also plotted a cross-sectional profile of specific target features in each modality showed improvement in contrast and sharpness (Fig. 5c). Strip-registration revealed fine details of dendrites of YFP-labeled retinal ganglion cells, vascular contrast, as well as higher contrast of nerve fiber bundles on the retinal surface (Fig. 5c). Additionally, strip-based motion correction improved the contrast and sharpness of the motion-contrast

that more accurately reveals perfusion of blood flow which is highly sensitive to motion artifact[27].

**AOSLO imaging reveals high-frequency, low-amplitude eye motion in awake, behaving mice.** In both the commercial SLO and custom AOSLO, we observed a rapid, but low-amplitude eye motion that caused single frames to exhibit intra-frame wobble and shear (Supplementary Video 4, 5). Such effects are often seen in scanning devices or those with "rolling-shutter" when motion exceeds the frame rate of the camera. The motion was not an artifact of the imaging system, as the effect was silenced upon the delivery of anesthesia (Supplementary Video 5). The eye motion in the direction orthogonal to the vessel can be quantified by tracking the shape of the shearing profile (Fig. 6). Using this approach, we quantified the rapid eye motion, that, to our knowledge, has not been observed in the awake mouse (Fig. 6a, b and Supplementary Video 8)[15,16]. A Fourier transform of the eye motion trace (Fig. 6c) revealed two prominent peaks at the low frequency, respectively, at 2 and 9 Hz. These peaks correspond to the characteristic respiratory[28] and cardiac frequencies[29] of the awake mouse. However, we also observed a band of power in the higher frequency regions over 150 Hz. Fourier transform of the eye motion velocity trace (Fig. 6d) shows a band of

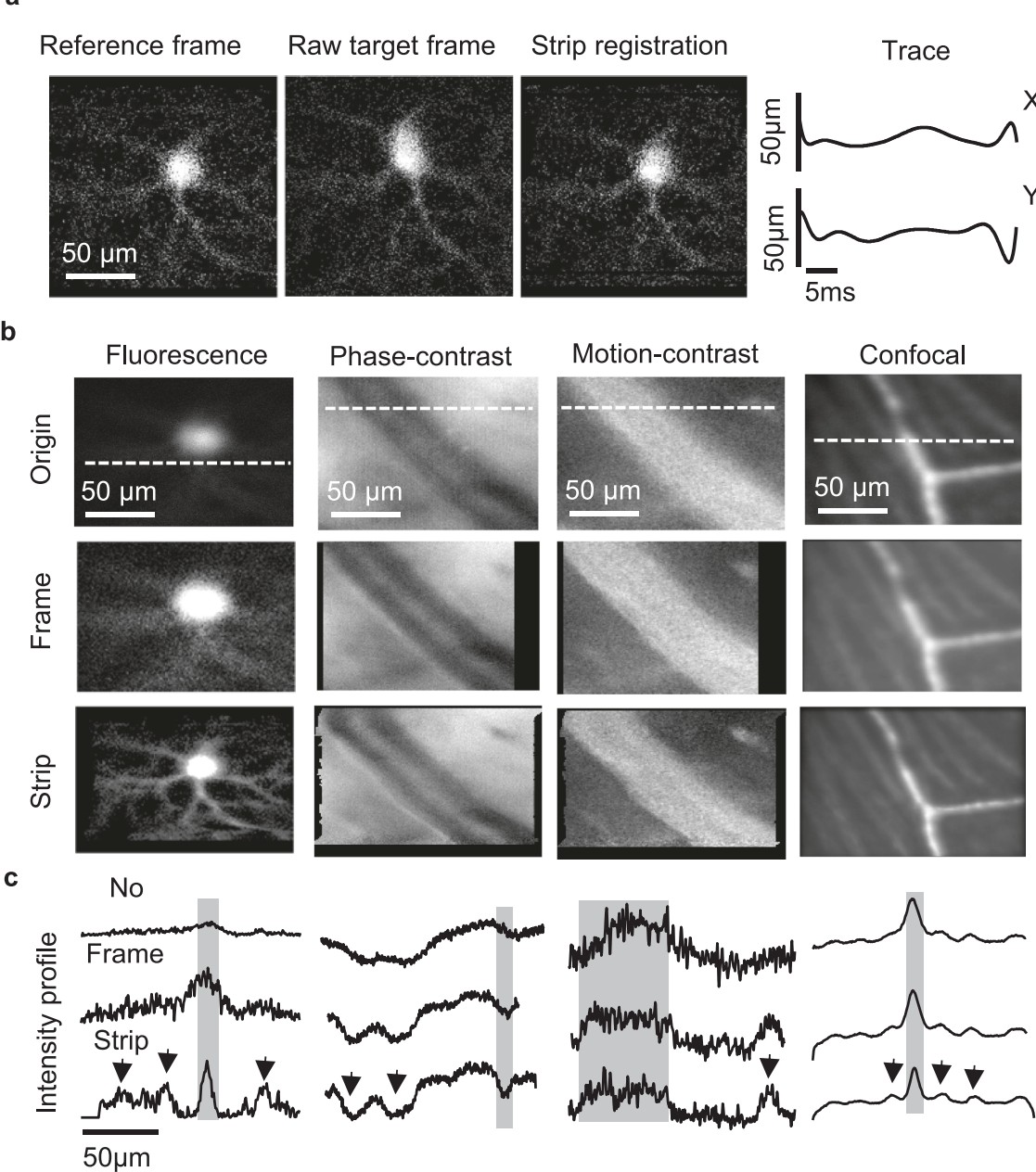

**Fig. 5 Strip registration for multimodal AOSLO imaging in the awake, behaving mice. a** Demonstration of strip registration using a YFP-labeled retinal ganglion cell imaged from a head-restrained awake mouse. Left to right: manually selected reference frame with the least distortion, raw AOSLO image distorted by the eye motions, registered image, and the eye motion trace on the horizontal and vertical directions. **b** Averaged images of four modalities with no registration (top), frame registration (middle), and strip registration (bottom). From left to right: Fluorescent image of a YFP-labeled retinal ganglion cell; phase-contrast image of a major retinal blood vessel (venule); motion-contrast image of the blood vessel; confocal reflectance image of capillaries and nerve fiber bundles. **c** The intensity profiles of three registration strategies which locations labeled as white dashed lines in (**b**). For comparison, the gray strips indicate the width of a main image features, while the arrows indicate other minor features. In **b**, **c**, strip registration shows improved image contrast and SNR in all four imaging modalities.

velocity near 100 Hz with bandwidth (30−150 Hz). In foveated species, this frequency band is characteristic of ocular tremor[30]; however, it is notable here as mice lack a fovea and have visual acuity of 0.5 c/deg[31]. High-frequency eye motion amplitude was ~6 μm which is far less than the resolution of conventional ophthalmic devices, which may explain why it has been missed previously.

**Long epoch imaging capability: behaving mouse imaged over 10 h.** Under anesthesia, it is rare that the mouse eye can be

imaged for more than 2 h at a time due to the tolerance of anesthesia, development of anterior opacification, or likelihood of recovery after sustained anesthesia[32–34]. Therefore, the awake preparation allows for semi-continuous imaging sessions that span many hours. To demonstrate this capacity and quality of imaging over long intervals, the same awake, behaving mouse was imaged for 10 consecutive hours with an interval of 1 h (Fig. 7a). The 10-h imaging time spanned both the normal wake and sleep periods of the circadian light-dark cycle (the first 4 h correspond to the light hours in vivarium, and the remaining 6 h corresponds

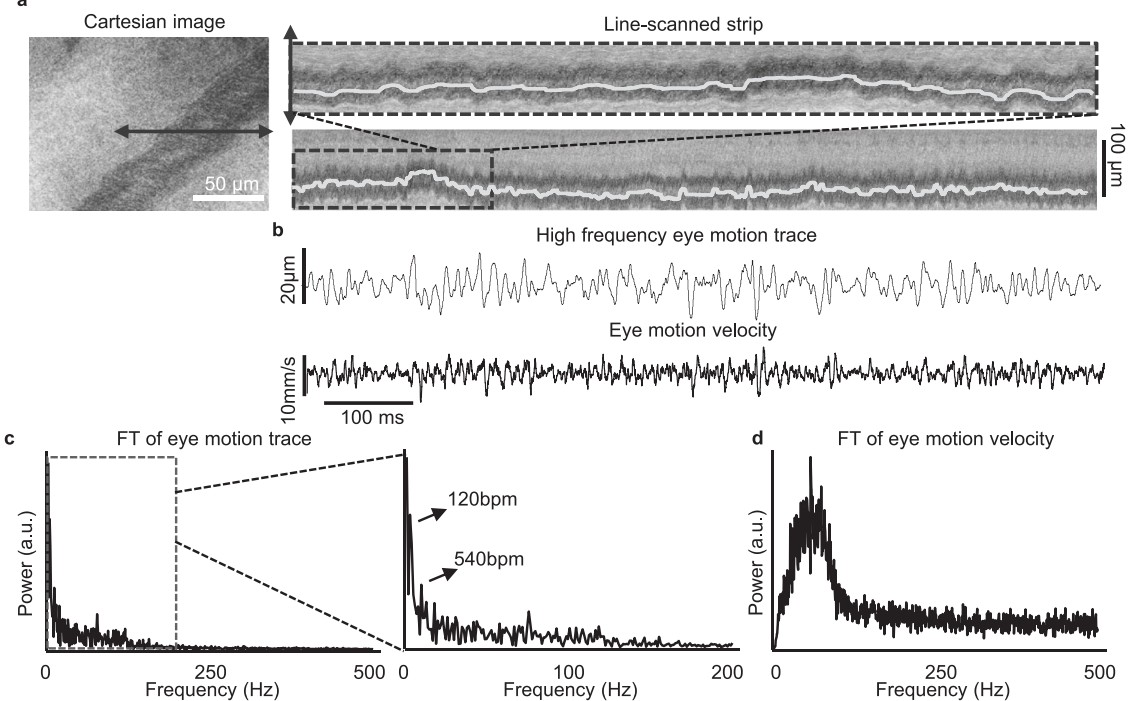

**Fig. 6 Measuring high-frequency eye tremor in the awake, behaving mice using line-scanned AOSLO. a** Measurement of low amplitude but high-frequency eye motions using line-scanned imaging across a major retinal blood vessel (vein). Left: Cartesian image of the vein. The black arrow indicates the placement of line-scanned imaging. Right: line-scanned image of the vein with motion trace overlaid as white line. **b** Top: Trace of high-frequency eye tremor highlighted using a high-pass filter (>30 Hz). Bottom: trace of the eye motion velocity. **c** Fourier transform of the eye motion trace (unfiltered). Power were observed up to 200 Hz. Two prominent low-frequency peaks were observed respectively at 2 Hz (120 bpm) and 9 Hz (540 bpm), which may be contributed by the respiratory and heartbeats. **d** Fourier transform of the eye motion velocity. Elevated power at 30–200 Hz were observed, indicating the bandwidth of the eye tremor.

to dark hours). Despite occasional gaze shifts that naturally change the imaging field, we could return to the same imaged field with a <3° accuracy by manually adjusting the orientation of the mouse to provide consistency over each imaging session (Fig. 7b and Supplementary Video 7). To show one such application, we studied the high-frequency eye motion as a function of the time of day spanning 10 h. The amplitude of the rapid eye motion was consistent over 10 h with a mean of 5.95 μm or 10.50 arc minutes (Fig. 7c). The power spectrums of the all-time point also suggested a consistent frequency bandwidth of the motion velocity (Fig. 7d, e and Supplementary Video 9).

**High-frequency eye motion removed by ketamine/xylazine anesthesia.** High-frequency eye motion was measured in one mouse first in the awake state, then after K/X injection for ~2 h. We observed that the high-frequency eye motion was rapidly removed after induction of anesthesia (Fig. 8a, b and Supplementary Video 10). Quantification of this effect is shown in Fig. 8; however, the visible attenuation of high-frequency motion is also apparent in Supplementary Video 5, where the same retinal locations are shown within minutes of K/X injection. The eye motion was <10 μm over 80 min, suggesting that not only gaze positional shifts are suppressed, but also high-frequency tremor-like motion was suppressed. Eighty minutes after the K/X injection, slight twitching and whisking of the face was observed. Both the fast and gaze-dependent eye motion gradually returned to baseline-like conditions (Fig. 8c).

**Retinal blood flow suppressed by ketamine/xylazine anesthesia.** Blood flow was measured by using the previously reported technique in the head-restrain mice[15]. Briefly, when performing line-scanned imaging across a blood vessel, the flowing blood cell generates an angled trajectory in the space-time image, indicating a change in position over time scaled by the angle of incidence of the beam across the vessel[15]. The velocity of the blood cells can be quantified by measuring the slope of blood cell profiles (Fig. 9a and Fig. S6). Flow was found to be pulsatile (Fig. 9b) but at different frequencies and velocities corresponding to variable heart rate with anesthesia (481 and 758 Hz in two awake mice), corresponding to the much higher cardiac frequencies[35] compared to that measured in anesthetized mice (271.2 ± 1.2 Hz, $n = 5$ blood vessels from three mice). Blood flow in a single venule was reduced by 43% relative to baseline measure (1.56 to 0.89 μL/min) 20 min after anesthesia (Fig. 9c). Independent measures showed an abrupt narrowing of the vessel diameter right after the K/X injection, while the change of blood velocity was minor at the beginning but substantially drops overtime after 20 min K/X injection. These two parameters did not show equivalent changes, which underscores the importance of direct measurements to reveal a complete picture of blood flow and its regulation[36]. While eye motion (discussed above) gradually returns 80 min after K/X induction, blood flow remained low (0.66 μL/min), which was 58% below baseline. These findings suggest the return of eye motion preceded the return of retinal blood flow, indicating the return of physiological function occurs at different rates in the same mouse after K/X anesthesia. Flow measurements from the awake mice ($n = 5$ vessels from three mice) were also compared to previously reported data from our group in anesthetized mice ($n = 123$ vessels from 20 mice, Fig. 9d)[15,19]. Despite single vessels where velocity was reduced under anesthesia, we find that overall, the flow vs vessel diameter relationship was generally similar. Fitted model diameter-flow

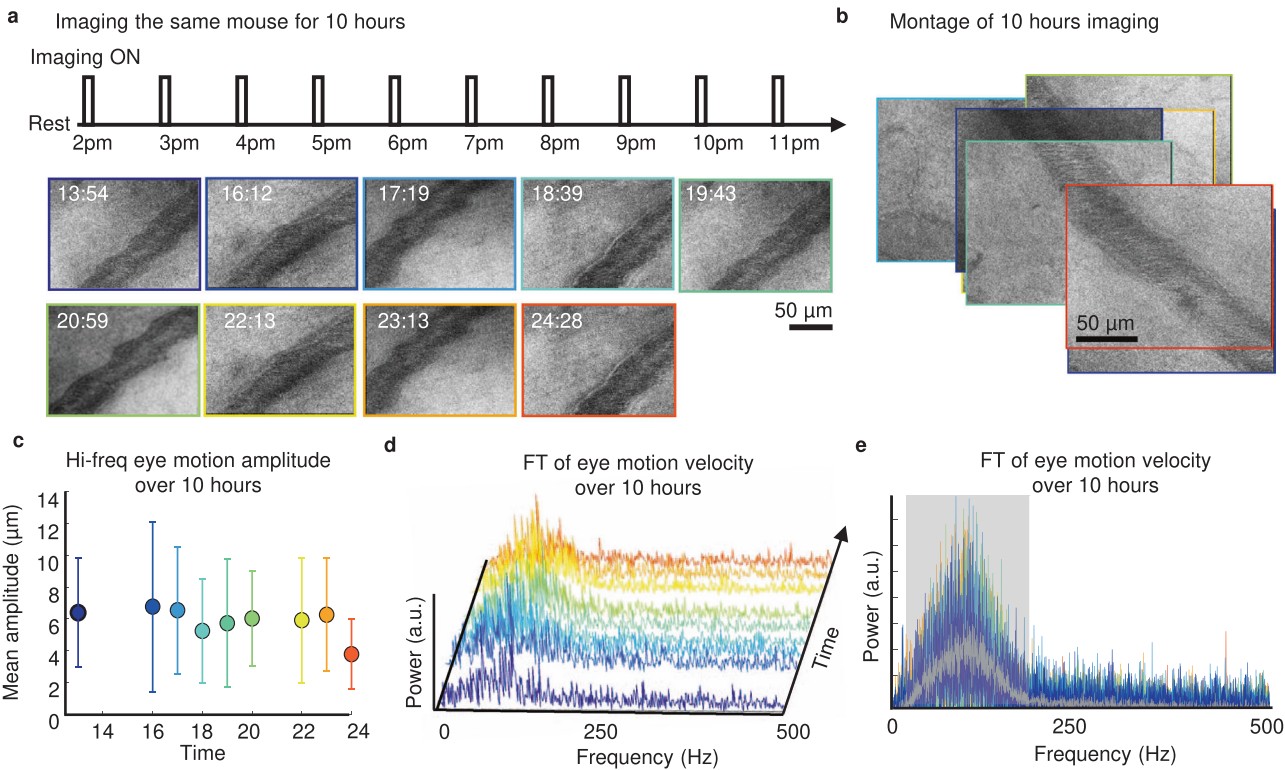

**Fig. 7 10 h longitudinal AOSLO imaging in a single mouse. a** 10 h longitudinal AOSLO imaging in the same head-restrained awake, behaving mouse. The imaging lasted for 11 h, with one 10-min imaging session for every 1–1.5 h. Representative Cartesian AOSLO images of the same vein at each session were shown. **b** Montage of Cartesian images with borders color-coded with timestamps shows the capability to navigate and localize to the same vein in different imaging sessions across 10 h. **c** Mean amplitude of the eye tremor measured from the line-scanned data of the same venule over 10 h. The error bar indicates the standard deviation of eye motion amplitude over all time points ($N = 15{,}000$ lines). **d** FT of eye motion velocity trace showing identical power spectra over 10 h with consistent bandwidth from 30–200 Hz. **e** Another perspective view of the same data as (**d**). Overlaid gray line shows the average of all power spectra.

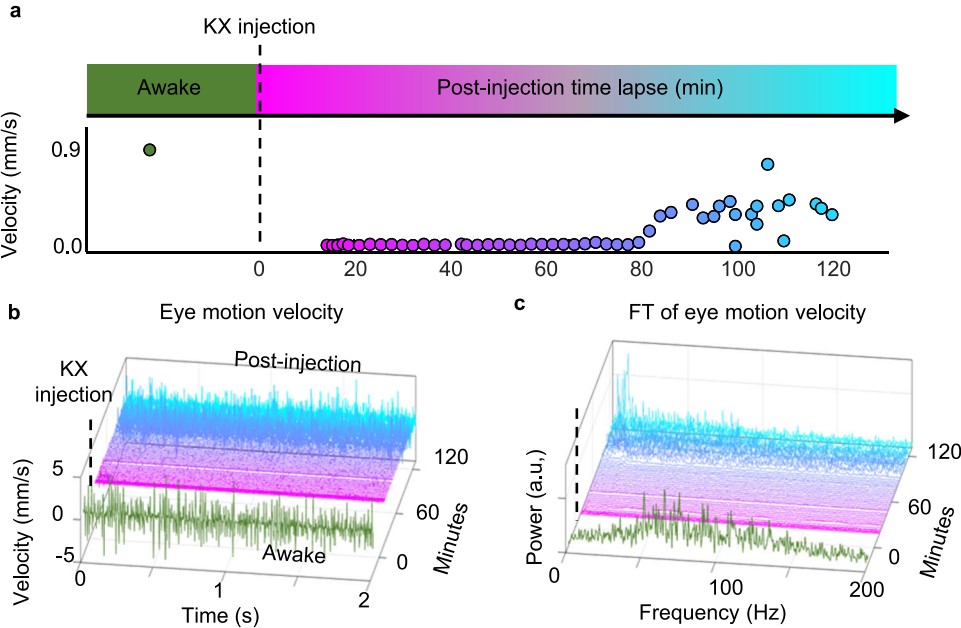

**Fig. 8 Effects of KX anesthesia on high-frequency, low-amplitude eye tremor. a** Measurement of eye motion shows the absence of eye motions under deep anesthesia, then reappears under shallow anesthesia but with lower velocity compared to the awake state. Color code indicates the time lapse of the imaging where green is the awake state, magenta is the deep anesthesia state, and cyan is the shallower anesthesia state. **b** Eye motion velocity profiles and **c** Its power spectra at each time point reveals relatively not only lower velocity but also lower frequency bandwidth under shallow anesthesia when compared to the awake state.

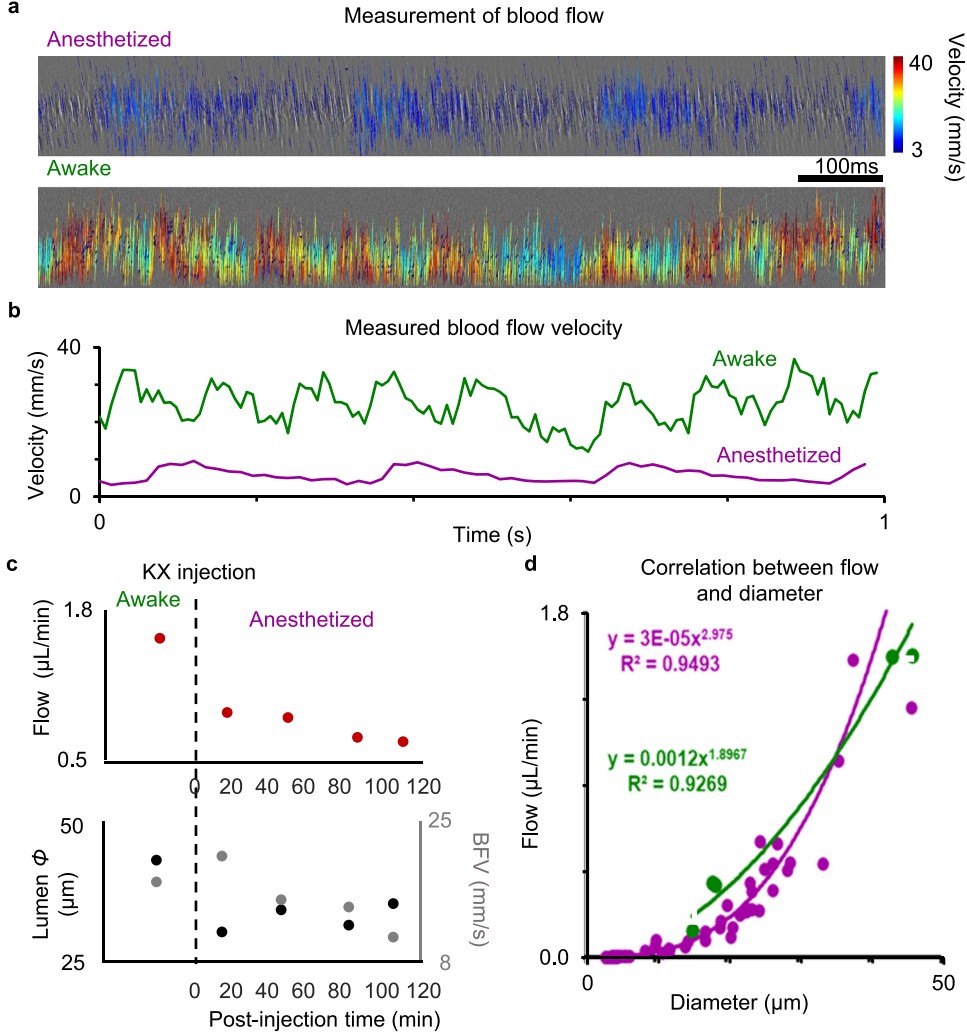

**Fig. 9 Effects of KX anesthesia on mouse retinal blood flow. a** Line-scanned image of the same large blood vessel in a mouse retina in the anesthesia (top) and awake (bottom) states. Detected flowing single blood cells were labeled with lines with color code represents their velocities. **b** Measured blood flow velocities of data shown in (**a**). Pulsatile was observed in both the awake and anesthesia data. Higher heart rate and flow velocity were observed in the awake mouse. **c** Measured blood flow in one vessel over time shows the suppression of blood flow under anesthesia. **d** Single-cell blood flow measured in awake mice (green, $n = 5$ vessel, 3 mice) are compared to a population of anesthetized mice (purple, $n = 123$ vessels, 20 mice). Flow-diameter model curves are fit to the data.

curves (detailed in Methods) show minor overall differences in total blood flow.

**Retinal thickening induced by ketamine/xylazine anesthesia.**
We measured the retinal morphology varied from awake to anesthetized state using OCT in four mice. All mice were imaged at the superior-temporal quadrant of the right eye with a 30° FOV. We observed that the mice subjected to anesthesia showed a substantial retinal thickening over two hours, while the mice in the awake states did not (Fig. 10a). Total retinal thickness (TRT) was quantified by a custom software from the 3D OCT cubes (Fig. 10b), measurements of each data point is shown in Fig. S8. At 80–110 min post-injection, retinal thickness in anesthetized mice reached a maximum increase of $8.0 \pm 4.1 \, \mu m$ ($n = 4$ mice) with an average $3.9 \pm 2.0\%$ thickening ($p = 0.034$, Fig. 10c), while no significant thickening was observed in retinas of awake control mice ($p = 0.179$, $n = 4$ mice). The thickening effect continued progress at the last time point of the measurement. *En face* map of TRT change also suggested such a thickening effect was uniform across retinal regions and eccentricities (Fig. 10d, e).

**Discussion**
The head-restrained mouse preparation facilitates a stable optical axis through which cellular-scale structural and functional measurements of the awake mouse eye can be imaged with standard and high-resolution ophthalmic imaging modalities. Creating this stable ocular imaging axis still allows for free mouse ambulation, which reduces confinement stress and further allows a gateway for behavioral studies and mouse visual interaction with environments with or without anesthesia. This enables a new domain of retinal research in the mouse that may examine and/or correct for the impact of anesthesia on retinal imaging, as many studies have examined such impact in the cortex[5]. To demonstrate one aspect of the impact of anesthesia, we have, for the first time, characterized and compared blood flow measurements in both the awake and anesthetized states in the same mice. By measuring retinal blood flow (RBF), we show that blood flow is suppressed by KX anesthesia. This confirms evidence from previous studies[9,37], but now provides micron-level resolution, which are critical for accurate blood flow measures that are contained within vessels far smaller than 3-45 microns in the mouse[15]. This work is essential for understanding the underpinnings of

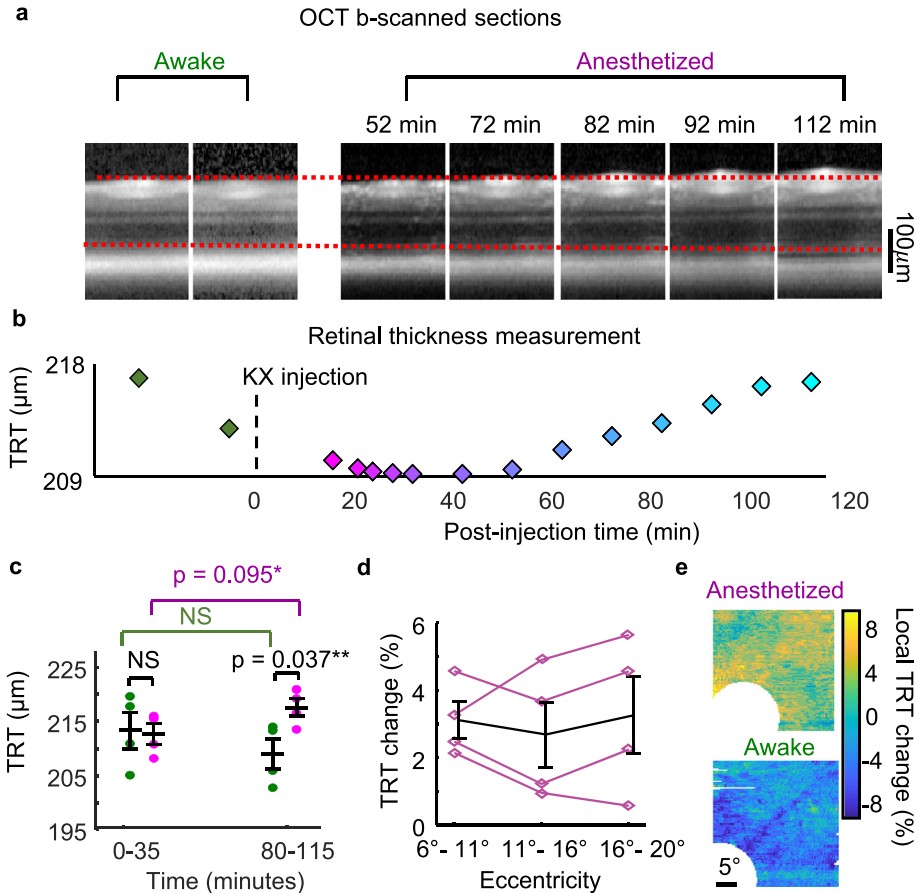

**Fig. 10 Effects of KX anesthesia on mouse retinal thickness. a** Longitudinal OCT mouse retinal imaging in the awake and anesthetized states. A retinal thickening was observed associated with post-injection time lapse of K/X anesthesia. **b** Total retinal thickness (TRT) measured from the OCT cubes. It shows a substantial thickening effect after 60 min post-injection time of KX anesthesia. **c** Measurements of $N = 4$ (Mean ± SD) mice shows significant thickening related to KX injection (two-sided student's *t*-test). **d** Evaluation of TRT changes versus three ranges of eccentricity on the retina show no significant difference (black line plot, Mean ± SD, $N = 4$ mice). **e** *En face* maps of TRT change also show no noticeable local difference related to regions or structures on the retina.

neurovascular coupling and regulation of microvascular flow in disease models that are best studied without the impact of anesthesia which may confound or blunt the dynamics and baseline state. We also find that KX anesthesia induces appreciable retinal thickening as measured by OCT. To our knowledge, such progressive thickening has not been reported, and the mechanisms are unclear. While speculative, we posit the observed retinal thickness and RBF change may be impacted by changes in intra-ocular pressure[38] which can be modulated by KX anesthesia[39]. KX anesthesia has also been found to induce respiratory and heart rate depression and decreases in mean arteriole pressure, which could induce a change of RBF[40]. While further work is needed to inform the nature of such changes, this finding underscores the importance of making such measurements in the awake state, as anesthesia may skew the interpretation of in vivo mouse measurements of disease, especially as it may confound conventional interpretations of retinal cell loss/survival.

It is worth reflecting that while in vivo retinal imaging studies have strived to remove eye motion with the goal of gaining a stable imaging platform, it comes with the consequence of removing natural input to the eye. Here, we recognize that leaving natural eye motion intact provides a chance to better understand how the eye motion and visual system works together to provide the organism vision[41]. To demonstrate one such example, we show active imaging during the free gaze behavior of the awake

mouse. Like humans, there is a temporal-nasal preference as the mouse scans the visual horizon[21]. We also find that mice have preferred gaze positions (Fig. 3) despite lacking fixation ability normally attributed to foveated mammals[21]. At a finer scale enabled by high spatiotemporal resolution AOSLO, we reveal a high-frequency, low-amplitude eye motion with a temporal frequency similar to the human ocular micro-tremor (OMT). Provocatively, the amplitude of the fast eye motion in both species corresponds roughly to half the size of the maximum reported visual acuity[30]. In further studies, it will be interesting to see if this ocular motion could be a conserved feature of mammalian vision that scales with receptive field size or visual acuity. To our knowledge, this is the first report of this high-frequency, low-amplitude motion in the mouse (Fig. 6). Incorporating this dynamic retinal visual input could have notable consequences for the spatiotemporal characterization of visual receptive fields in the mouse that span the retina-to-cortex[42-44]. We show that such eye motion is maintained throughout 10 h across the day/night circadian cycle (Fig. 7). Importantly, this high-frequency eye motion is eliminated with the application of anesthesia, which is beneficial for high-resolution imaging (eliminating motion blur); however, also removes this information as natural input to video-rate or static visual stimuli. Further investigation may reveal a common requirement of mammalian vision to incorporate purposeful eye motion at this microscopic scale to improve spatiotemporal contrast for the visual system[30,41]. Aligned with this

interest, the impact of gait has, in some studies, indicated an apparent increase in visual acuity and response gain of certain visual neurons associated with locomotion in the mouse[45] which may now be studied at the retinal level. We also consider that the measurements of such low amplitude, high-frequency eye motion could provide a new, useful biomarker for neurodegenerative diseases in numerous mouse models that seek to quantify the integrity of neural circuits responsible for eye motion control[46]. Finally, it is worth noting that beyond the study of such eye motion in the pursuit of understanding neural function and control, accounting and correcting for eye motion is essential in achieving high-resolution images of the retina. Left uncorrected, it will induce motion blur in high-resolution imaging modalities with exposure times more than ~5 ms (200 Hz frame rate). And while flash photography could mitigate such blur, there are few video acquisition ophthalmoscopes to date that achieve these imaging speeds, especially for the mouse. This would mean that even through perfect optics, the awake retinal image would be blurred on a scale of 10–20 micrometers. Therefore, high-resolution imaging needs to achieve both aberration correction and high-speed imaging/registration >200 Hz to achieve diffraction-limited potential in the awake mouse.

Forgoing anesthesia also has a major advantage in studying in vivo physiology over longer time intervals. Studies to date in the mouse have focused on anesthetized retinal imaging that is largely constrained to 30–120 min. This is due to anesthesia intolerance (poor animal survival) or transient cataract formation when imaging for longer periods. Here we show that it is possible to perform a semi-continuous imaging protocol that allows imaging of the same animal for over 10 h or longer. Fully continuous recordings are possible if the right scientific questions warrant; however, breaks for animals to recover in their cage to eat, rest, and sleep are recommended. This paradigm opens an enormous temporal window to finely probe physiological challenges at the scale of hours, if not days. For example, circadian changes, response to drug therapy, immune cell movement[36], systemic glycemic modulation, and exercise regimes[47] can now be studied over the course seconds-to-hours in a single session, and semi-continuously over days without accumulated anesthesia toxicity or complications of anterior opacities that have classically limited such study requiring anesthesia.

The laboratory mouse is among the most popular models used to recapitulate aspects of human disease, and here we move the model another step closer to capture the normal physiology of the mammalian eye by obviating the need for anesthesia. Toward making this approach globally available, a low-cost 3D-printable solution is provided that can be replicated in any lab that has access to a 3D printer and access to free-market parts. This design is intended to provide proof-of-concept and democratize stable ocular measurements in the mouse without anesthesia. There are myriad applications that may build on this work that have the potential to advance our understanding of the fundamentals of vision, behavior, neural structure, functional integrity, and basic physiology of mouse vision. We expect some of the earliest applications will probe the impact of anesthesia on a number of basic physiological functions. Others may use the resolution and retinal tracking capabilities to improve fast and microscopic eye position recordings in free-viewing or simulated visual environments. We contend that comparisons of visual physiology between mouse and man is now even closer by obviating the need for anesthesia which is rarely used in clinical ophthalmic imaging. Finally, the value of long-term evaluation lasting hours-to-days will enable a new regime of the study of physiology and structure across a new window of time that is critically valuable for response to therapy, circadian cycles, and more gradual changes that would otherwise be missed due to the logistical challenges of limited imaging windows confined by anesthesia. Combined, we

expect this to represent the beginning of exciting new studies in neural and behavioral studies in the mouse, one of the most powerful partners in biomedical research.

## Methods

**Head-restrained setup for awake-behaving mouse retinal imaging**. The mouse head was restrained by a clamp holding a laterally positioned arm of its surgically implanted Y-shaped headplate. The headplate was positioned vertically above a running wheel (90 mm diameter) such that the mouse can ambulate freely with a natural posture and gait during imaging. Lateral positioning of the clamp also allows for feasible imaging of the contralateral eye. The clamp consisted of two interlocking parts 3D printed to create a tight fit around the headplate, ensuring stable head restraint despite only being held on one side. All components of the wheel platform, except the bearing and axel, were 3D printed with polylactic acids (Ultimaker, Ultimaker B.V., Utrecht, Netherlands). A rotary encoder (ENS1J-B28 L00256L, Bourns, Inc, Riverside, CA, USA) was attached to the wheel axle, allowing them to turn in sync. The encoder output was acquired by a data acquisition board (USB-6001, National Instruments, Austin, TX, USA). The gait distance and speed were extracted from the encoder output signals with MATLAB R2019a (The MathWorks, Inc., Massachusetts, USA). The locomotion data was synchronized offline based on the timestamps of the simultaneously acquired imaging data. The 3D printing files and detailed assembly instructions are accessible publicly at "https://github.com/GP-Feng/Awake_Mouse_Imaging_CommBio.git".

**Animals**. All animal studies were performed as per the guidelines of the National Institute of Health and approved by the University Committee on Animal Resources at the University of Rochester. C57BL/6J mice were purchased from The Jackson Laboratories (Bar Harbor, ME, USA). Twelve mice at 6–10 months old were used for experiments. All animals were housed in The Vivarium laboratory animal facility at the University of Rochester Medical Center on a 12-and-12 h light-dark cycle with food and water ad libitum.

**Headplate implantation and animal training**. The timeline protocol of headplate implantation and animal training is shown in Fig. S1. A subcutaneous buprenorphine injection (0.5–1 mg/kg) was administered 24 h prior to surgery. On the day of surgery, mice were weighed, anesthetized, transferred to a small animal stereotactic frame (900 series model, David Kopf Instruments, Tujunga, CA, USA), and secured with the ear bars and incisor bar set for a flat skull. The dorsal surface of the head was first shaved, and the skin was then cleaned with a 70% isopropyl alcohol prep pad (Catalog number: MDS090735, Medline, Northfield, IL, USA). A mid-line incision was made from the base of the skull to the frontal bone between the eyes, approximately 2–3 cm in length. Then lateral incisions were made to where the temporal muscle inserts into the skull. The overlying fascia was gently removed by blunt dissection and frequent irrigation to expose the skull surface. A Y-shaped 3D printed headplate was then held firmly on the skull by applying a mixture of dental cement (Jet XR Shadow Powder, Lang Dental, Wheeling, IL, USA) and cyanoacrylate glue (2:1 ratio) (Krazy Glue Maximum Bond, Krazy Glue, USA). The headplate should be centered over the intersection of the coronal suture and bregma, with one medial arm aligned with the midline towards the nasal bone. Mice were put in a single housing with cardboard houses to recover for 3 days post-surgery. Overall, mouse health was monitored every day by visual observation of hair coat appearance and activity level. Following recovery, at least five training sessions of 30–60 min were performed to acclimatize mice to the head-restrained imaging platform. Mouse weight was monitored for each training session. In the first two sessions, mice were shallowly anesthetized with a ketamine-xylazine mixture (100 mg/kg ketamine and 10 mg/kg xylazine) and then recovered with head-restrained on the platform to allow for familiarization. After recovery, the mouse was allowed to ambulate for 30-45 minutes. These two training sessions were performed 24 h apart to avoid consecutive application of anesthesia. Three or more subsequent 45-minute training sessions without any anesthesia were conducted, allowing mice to acclimatize to the head restraint condition in the fully awake state. Mice that display steady gait and posture, as well as relaxed behavioral characteristics such as grooming, are considered acclimatized and ready for imaging.

**Imaging protocols**. When imaging with the awake-behaving mouse, no additional procedures were applied in the eye except pupil dilation. Pupil dilation was applied in either the awake or anesthetized mice, which was achieved by putting in eye drops of 1% tropicamide (Sandoz, Basel, Switzerland) and 2.5% phenylephrine (Akorn, Lake Forest, Illinois, USA). Imaging was performed in a dark environment with a 10–15 min dark adaptation phase prior to the start of each imaging session. Because the adjustment of the roll rotation axis with the mouse was limited in our platform, the FOV of AOSLO imaging were mostly at the upper quadrant of the retina. When imaging with anesthetized mice, intraperitoneal injection with a mixture of ketamine hydrochloride (100 mg/kg dosage) and xylazine (10 mg/kg dosage) was applied. An external heating pad and rectal probe thermometer (Physiosuite, Kent) was used to monitor and maintain a body temperature of 37.0 °C for the anesthetized mice. To prevent ocular opacification induced by dehydration under anesthesia, a contact lens (Advanced Vision Technologies,

Lakewood, Colorado, USA) was placed over the eye being imaged, and an artificial tear (GenTeal, Alcon Laboratories, Inc, Fort Worth, Texas, USA) was routinely applied every 20 minutes during imaging.

**Instrumentation.** SLO and OCT imaging were performed with a commercial multimodal retinal imaging platform (HRA Spectralis, Heidelberg Engineering, Heidelberg, Germany). Both OCT and SLO images were captured with the 30° FOV objective unless being specified. The pixel resolution of SLO was $1536 \times 1536$, while the OCT cubes had $1536 \times 596 \times 97$ voxels. The *en face* FOV of the OCT cube was $20° \times 20°$ and with registration using the built-in Automated-retinal-tracking (ART) function. Each OCT slice was also averaged for 40 frames to improve SNR.

The multi-channel AOSLO was described in the previous literature[14]. It permits retinal imaging with modalities including reflectance confocal, phase-contrast, and fluorescent imaging. Confocal reflectance and phase-contrast imaging was performed with a 796 nm superluminescent laser diode with 17 nm bandwidth (200–500 μW at the cornea, S790-G-I-15, Superlum, Ireland), while a 488 nm laser diode (220–330 μW at the cornea, iChrome MLE, Toptica Photonics, Farmington, NY, USA) was used as the excitation source for fluorescent imaging. A 2.1 ADD pinhole was used in the image plane of the NIR channel for confocal reflectance and phase-contrast imaging. To provide phase-contrast imaging, the pinhole was offset laterally for ~22 ADD (Guevara, 2020). A Hartmann–Shack wavefront sensor (HSWS) was applied to measure the wavefront of the mouse eye in real-time. Combined with the deformable mirror (DM97-1, ALPAO, Montbonnot-Saint-Martin, France), the inherent aberration in the mouse eye was corrected in close-loop.

**Pupil matching measurement.** By focusing on the eyelids and canthus, the exit pupil of the mouse eye was imaged using NIR SLO in a speed of 8.8 fps with $768 \times 768$ pixels. Imaging was conducted in five healthy headplate mice in the awake state. Each imaging session spanned ~20 min, and 17 videos were recorded for the maximum length that the system allowed (59.88 s), with gaps varied from 5–30 s depending on the data buffering time between two acquisitions. A custom-built pupil tracking algorithm written in Matlab was used to automatically track the pupil region from the SLO video. A texture-based segmentation strategy was used (Fig. S2). First, a standard deviation (STD) filter was applied to the image to highlight the pupil region, whereas the pupil has a lower spatial variance compared to the surrounding tissues, such as the eyelid and fur. The spatial STD map was then binarized with thresholding (STD <0.25) and the pupil region was segmented with a watershed algorithm. The contour of the pupil area was finally fitted with an elliptical function. Logic gates were set to remove false positives with abnormally large and rapid changes in the detected pupil size and position. The pupil matching performance was quantified as the overlapping area between the segmented exit pupil of the mouse eye and a static virtual circular entrance pupil located at the mean position of all data points. When analyzing the correlation with the mouse locomotion, the pupil matching data were averaged across 1 s time bins to align the simultaneously recorded locomotion data.

**Retinal tracking and blink detection for SLO imaging.** The experiment protocols for blinking and gaze shift measurement was identical to the pupil measurement, which the only difference is the SLO was focused on the nerve fiber layer in the retina. Here, the blinking event was detected based on the mean image intensity of the frames. A frame was considered to experience a blinking event when its image intensity is below 60% average image intensity of the whole video (Fig. S3). We implemented a semi-automated-retinal tracking algorithm written in Matlab to quantify global offsets for the SLO retinal image. Three different reference templates with well-defined features, such as the optic disc and blood vessel junctions, were first manually selected from a reference image (Fig. S4a). The size and locations of the templates were determined by experienced users through a guided user interface. The three templates were used as a reference to perform 2D cross-correlation with each retinal image frame (Fig. S4b). The retinal offset was determined by one of the three relative offset extracted from the 2D cross-correlation, which has the largest normalized correlation coefficient (NCC) (Fig. S4c). Logit gates was implemented to remove date points when experiencing blinks.

**AOSLO motion correction.** A previously reported custom strip-based registration algorithm was used to correct the motion distortion for AOSLO images[26]. Briefly, the image was separated into strips along the slow scanning axis, and each strip was registered to the reference image individually with 2D cross-correlation to correct the shearing effect induced by the fast eye motions. Here, the width of the strip selected was 32 lines (equivalent to a sampling rate of 468 Hz) which is slightly over the Nyquist rate of the upper bandwidth of the high-frequency eye motions (~200 Hz). Registered strips with NCC smaller than 0.5 were considered "failure registration" and removed from the registered image. To register the fluorescent image, all images was being convolved with a 2D Gaussian kernel with $\sigma = 5$ pixels prior to registration to improve the 2D cross-correlation performance for weak photon counts.

**Miniature eye motion measurement.** The high-frequency, low-amplitude eye motion was measured with AOSLO by scanning a 1D imaging beam across a major retinal blood vessel at 15 kHz. When the eye motion exists, the imaged spatio-temporal profile of the blood vessel will be sheared (Fig. S5a right). The amplitude of the eye motion component orthogonal to the blood vessel $|w_I|$ can be extracted as: $|w_I| = |v|\sin(\theta)$, where $\theta$ is the angle between the blood vessel and the scanning axis, and $|v|$ is the shear of space-time blood vessel profile (Fig. S5a left). A fully automated algorithm was used to extract the shear trace similar to the strip-registration algorithm for motion correction, but it only performs 1D cross-correlation along the spatial direction. Twenty reference strips with width of 32 scanning lines were first randomly selected from the spatiotemporal image (Fig. S5b). A 1D cross-correlation was conducted along the space axis with each reference-strip to produce 20 candidate traces, which were then aligned by normalizing to the mean values. Offset data points in the candidate traces have changes larger than 10 pixels compared to the previous time point, were treated as false positives and were excluded from the future analysis. Finally, the 20 candidate traces were merged as the final output motion trace point-by-point by calculating the median value (Fig. S5c). The extracted motion traces were validated by visual examination performed by an experienced user by investigating the registered space-time images (Fig. S5d), measurements with badly registered image were considered "fail" and were removed from the analysis.

**Blood flow measurement.** Single-cell blood flow was measured noninvasively in awake mice using an approach which has been described extensively in our previous publication[15]. Briefly, blood vessels were obliquely scanned in 1D (15 kHz) by freezing the slow (galvo) scanner and consecutive scans were concatenated through time, to form a space-time image (Fig. S6a). Blood cells appear as diagonal streaks, indicating their position across the scan. The slope of the streak combined with the angle between the direction of flow and the position of the 1D scan (extracted from subsequently acquired 2D raster image) were used in an automated algorithm employing the Radon transform to extract blood cell velocity. In our space-time images, streaks approaching a vertical orientation represent higher velocity cells. The radon algorithm operates by splitting space-time images into overlapping ROI's. Each ROI is iteratively rotated 180° and a 1D intensity projection is extracted for each iteration. The standard deviation was calculated for each 1D projection and the highest value indicated the dominant angle and thus yielding a velocity for the ROI. The measurement bandwidth of this technique was 0.03–1275 mm/s, which was more than sufficient to measure biologically possible blood cell speeds in the awake-behaving mouse retina. The pulsatile flow was measured with high spatiotemporal resolution and overlaid as a velocity color map on the space-time image (Fig. S6b). To determine vessel diameter, motion-contrast images of the blood vessel were generated from 2D Cartesian images. The imaging of forward and multiply-scattered light through the RBC column within microvessels spanning <50 μm in size helped obtain an accurate measurement of the lumen diameter. Combined, the velocity and diameter measurements gave the mean flow rate (in μL/min) through each vessel. These measurements were done simultaneously with those of eye movements described in the section above. Flow-diameter model curves were fit to the measured data compared to the two populations. Briefly, the model used was of the form $y = ax^b$, where y is the flow rate (μL/min) and x is the diameter (μm).

**Longitudinal retinal thickness measurement.** Retinal thickness measurement was conducted with HRA OCT. "Follow-up" mode was used to perform longitudinal imaging at the same location in the retina with the first time point as reference. The total retinal thickness (TRT) was defined as the distance between the vitreous-internal limiting membrane (vitreous-ILM) boundary and the outer segment-retinal pigment epithelium (OS-RPE) boundary (Fig. S7a, b). A circular region centered at the optic disc with a visual degree of 8°, which is difficult for segmentation, was masked out in the analysis (Fig. S7c). The layers were detected by a custom graph-theory-dynamic programming segmentation algorithm based on the strategy previously reported by ref. [48].

**Statistics and reproducibility.** All data in the current study came from 12 mice on the C57BL/6 J background from Jackson Labs (Bar Harbor, ME) stock (CX3Cr1, thy-1 YFP, B6 stock). Mice were either imaged as direct from Jackson labs or in a few generations of colony progeny from the original founder mice. All measurements in the study were performed either multiple times in the same mouse (e.g., Figs. 2–4, 6–9) or across multiple mice for comparison. When comparing across two different times or groups, differences were evaluated based on a simple paired t-test to compare conditions. Statistical tests were performed in Matlab® and reported with the threshold of significance meeting p values $*p < 0.1$, $**p < 0.05$. Actual p values are reported.

**Reporting summary.** Further information on research design is available in the Nature Portfolio Reporting Summary linked to this article.

## Data availability

Source data behind the figures are included in Supplementary Data 1–7 files. Processed SLO data and raw AOSLO and OCT videos are available at a public repository (Zenodo):

https://doi.org/10.5281/zenodo.7806898. Additional data is available from the authors on reasonable request.

## Code availability

3D printing files of the head-restrained mouse setup and all codes for data processing are available at the public repository (GitHub): https://github.com/GP-Feng/Awake_Mouse_Imaging_CommBio.

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

## Acknowledgements

Schallek laboratory is supported by the National Eye Institute of the National Institutes of Health under R01 EY028293 and P30 EY001319. Padmanabhan is supported by grants from the National Institutes of Health R01 MH113924, P50 HD103536, and National Science Foundation (NSF) CAREER 1749772. The research was also supported by an Unrestricted Grant to the University of Rochester Department of Ophthalmology, a Career Development Award, a Career Advancement Award, and Stein Award from Research to Prevent Blindness (RPB), New York, New York; the Dana Foundation David Mahoney Neuroimaging Award and a research grant from Genentech Inc. (Schallek).

## Author contributions

J.S. conceived this study. K.P. provided the idea and design of the head-restrained mouse preparation and setup. G.F., A.J., K.D., F.S., C.W.P., D.P., and J.S. designed and performed experiments. G.F., A.J., and F.S. analyzed and interpreted the data. G.F., A.J., K.D., F.S., D.P., and J.S. wrote the manuscript. All authors proofread and edited the manuscript.

## Competing interests

The laboratory was supported in part by a collaborative research grant from Genentech Inc. Jesse Schallek holds US patents on ophthalmic imaging technology held through the University of Rochester. The remaining authors declare no competing interests.
