## [Peer Review File · Communications Biology]

Reviewers' comments:

Reviewer #1 (Remarks to the Author):

In this manuscript authors have developed a new model to study physiology of mouse eye without the use of anesthesia. They proposed a head-restrained preparation with a stable optical axis that facilitates high-resolution structural and functional retinal imaging. They have imagined a 3D printed apparatus to hold the mouse head with body suspended above a rotational cylinder that allows free ambulation. They have validated that that image quality in awake animals are in accordance with images with a commercial scanning laser ophthalmoscopy. Then they study the impact of anesthesia on eye motion, blood flow and retinal thickness. They show differences between these parameters between anesthetized and awake animals.

The paper is well written and easy to follow. The model is interesting and videos illustrate well the data. The study is clearly laid out with convincing results of the use of this model for specific applications. However, the system for head retention need to be more detailed and illustrated. For example, a figure showing more precisely the headplate implantation and a video showing the animal with the system will help to understand the preparation.

I have some reserves on the benefit of the method to study visual system given that there are stages of surgery and training of the animal. These are important procedures for the animal with risks before imaging procedures. Can the authors discuss the real benefit of this method for studying retinal or optic nerve pathologies because functional exploration of vision in mouse can be done without surgery?

Moreover, this protocol may induce stress and stress can have profound effects on both animal's behavior as well as physiology. Can author discuss about how stress might factor into awake-behaving animal experiments that is important for interpreting results and making valid conclusions?

The proposed study is interesting but can be used for specific applications with regard to the technical complexity of the system to be set up.

Nevertheless, the idea developed in this manuscript is interesting and should be shared with the scientific community.

Reviewer #2 (Remarks to the Author):

This manuscript is, to my knowledge, the first to accomplish imaging of mouse retinae in an awake, mobile state. This is important. The authors make 3 significant claims, that high freq low amplitude eye movements are present in the awake mouse not present (due to anesthesia) or unnoticed by prior observers, that retinal blood flow is reduced in mice under anesthesia, and that mouse retinae thicken with anesthesia (all comparisons are between awake and KX anesthesia). They provide extensive data in support of these statements though many of the figures are quite crowded and difficult to read. I know these researchers and have significant confidence in their data as presented but have some issues with either their conclusions or the significance of those conclusions. Numerous relevant comments are made in the manuscript, unfortunately many about the English and I do feel the manuscript should have arrived in a more polished state with more tempered language about their results.

With regard to the first result on mouse eye movements this new high frequency component is of the order of 6 microns on the retina and I feel appropriate and sophisticated measurements were done to quantify these eye movements. As far as I could determine from the relevant literature, the mouse retinal ganglion cell receptive fields are much larger than this. I believe an ocular movement has to be on the order of 1/2 receptive field in size to likely be of physiological import and these movements are much smaller than that. So they exist and are now newly reported but I personally doubt their physiological significance.

With regard to retinal blood flow being specifically reduced by anesthesia, I don't doubt that it is given the large decrease in pulse rate and blood pressure. I would ideally like to have addressed to what degree a 'purely passive' system would have had decreased blood flow given the reduction in

the relevant cardiovascular parameters induced by anesthesia. Also I feel the description of the calculation of the blood flow was deficient given the great detail into which the paper went on how blood velocity and vascular diameter were measured. Again, all well done at the highest level of current retinal imaging technology.

With regard to mouse retinae thickening with anesthesia, there does seem to be an effect largely seen in mice beyond 80 minutes of anesthesia but the effect is not terribly large though it does seem to be statistically significant. So this may be important in sequential measurements of mouse retinal thickness if an experiment is done under anesthesia and is carried out for extended periods of time but there seems to be little effect of lesser durations of anesthesia.

So I think this paper contains much carefully obtained data on imaging of the awake mouse but I would be much happier with it if it were more self critical.

Reviewer #1 (Remarks to the Author):

The paper is well written and easy to follow. The model is interesting and videos illustrate well the data. The study is clearly laid out with convincing results of the use of this model for specific applications. However, the system for head retention need to be more detailed and illustrated. For example, a figure showing more precisely the headplate implantation and a video showing the animal with the system will help to understand the preparation.

Author Response: Thank you for the complements on the writing style and positive feedback on our manuscript. We have added additional panels in Figure S1 to complement figure 1 to show more precisely the head retention setup. In the additional figures, we provided: 1. A schematic to illustrate the positioning of the headplate on the mouse skull. Due to some objections regarding surgical photographs, we have opted to use a cartoon instead of a photograph. 2. After surgery, a photo of the headplated mouse on the wheel shows the apparatus of the mouse with its head motion constrained. To illustrate imaging and mouse behavior in the setup, we also show an eye-focused video in Supplementary Video1 which reveals the mouse motion as well as a zoom into the mouse eye which demonstrates the stability (and remaining eye motion) of the preparation. We thank the reviewer for this suggestion and we think it strengthens the basic concept.

I have some reserves on the benefit of the method to study visual system given that there are stages of surgery and training of the animal. These are important procedures for the animal with risks before imaging procedures. Can the authors discuss the real benefit of this method for studying retinal or optic nerve pathologies because functional exploration of vision in mouse can be done without surgery?

Author Response: We agree that functional exploration of vision in mouse has been previously accomplished with anesthesia and often without surgery. However, this is the major advance of our report, removing the confound of anesthesia has enormous potential to unravel the underpinnings of mouse CNS and retinal physiology without altered neural/vascular dynamics. This is a major field of study in the CNS literature that studies cortex function.

The benefit of our method is to realize high-resolution imaging to study the retinal physiologies and pathologies without confound from the anesthetics. In regard to the surgical approach for stability, we agree that future refinements may forgo the use of a headplate, however, this is commonly performed in a vast literature of awake-behaving vision studies in the mouse cortex. Therefore, it lends to a similar, if not identical platform through which cortex and retina behavior can be understood.

Finally, while this report requires surgery and training, the risk of distress and infection is minimal. In our study, all 5 of 5 mice survived the surgery. The incision recovered without infection within 3 days and ready for training. Mice resided in the normal cages with other co-habited mice for months. All animals were observed healthy without any noticeable signs of side effects related to implantation. Details of surgery and training were discussed below.

"The general approach and setup is presented in Fig. 1a. The mouse cranium was held stable through an affixed headplate. The Y-shaped headplate was implanted on the center of the mouse cranium in parallel to the anterior-posterior axis after a single surgical cut through the scalp under anesthesia (Fig. 1b). All mice survived the procedure. Headplated mice exhibited normal behavior and maintained a healthy outward appearance with groomed hair coat and normal activity level. No inflammation or other side effects related to the headplate implantation were observed. Weights of all mice also remained consistent pre- and post-surgery. Within 3 days of the surgical placement of the headplate, head-restrained mice became acclimated to the apparatus over 5 training sessions lasting 30-60 minutes. The Y-shaped headplate was mounted to a fixed arm suspended over a rotational cylinder such that the mice were able to freely ambulate fore-and-aft. After training and acclimation in the head fixed apparatus, mice did not exhibit aversion or retreat response maintained normal grooming behavior (Supplementary video 1) further indicating a calm baseline state with no outward signs of distress^{17,18}. The headplate surgery left the eye open with normal blink reflex (Supplementary video 1)."

Moreover, this protocol may induce stress and stress can have profound effects on both animal's behavior as well as physiology. Can author discuss about how stress might factor into awake-behaving animal experiments that is important for interpreting results and making valid conclusions?

Author Response: To provide more description about the stress level of the animals, Supplementary Video 1 shows the grooming behavior of a mouse during imaging. By visual outward appearance, there is normal grooming and behavior of the mouse. There is no wincing, lethargic, barbering (chewing) or hyperactive behavior based on

the distress scale provided by Paster et al 2009. We have modified the text in the manuscript to read:

"After training and acclimation in the head fixed apparatus, mice did not exhibit aversion or retreat response maintained normal grooming behavior (Supplementary video 1) further indicating a calm baseline state with no outward signs of distress^{17,18}. The headplate surgery left the eye open with normal blink reflex (Supplementary video 1)."

Combined, this indicates the stress level of the mouse is low. Moreover, spread over multiple days, established training session acclimate mice to the wheel apparatus, which allows free ambulation. We agree that stress has the ability to confound results and we have taken every step possible to mitigate animal stress. We emphasize that this same head-plated apparatus is nearly equivalent to similar studies in the cortex that contend with the same training and constrained regimes we employ to stabilize the head and eye.

Reviewer #2 (Remarks to the Author):

This manuscript is, to my knowledge, the first to accomplish imaging of mouse retinae in an awake, mobile state. This is important. The authors make 3 significant claims, that high freq low amplitude eye movements are present in the awake mouse not present (due to anesthesia) or unnoticed by prior observers, that retinal blood flow is reduced in mice under anesthesia, and that mouse retinae thicken with anesthesia (all comparisons are between awake and KX anesthesia). They provide extensive data in support of these statements though many of the figures are quite crowded and difficult to read.

Author Response: Thank you for the comments and underscoring the impact and relevance of our manuscript.

We fully agree that figures are quite compact in the prior version. Here, we have divided up key figures to address more focused points. Conforming with the journal's style criteria, we now have 10 figures (as opposed to 6 in the prior version). This allows for a more relaxed viewing and less crowding than in the prior version. Additionally, where appropriate, we have added multiple supplement videos and figures.

I know these researchers and have significant confidence in their data as presented but have some issues with either their conclusions or the significance of those conclusions. Numerous relevant comments are made in the manuscript, unfortunately many about the English and I do feel the manuscript

should have arrived in a more polished state with more tempered language about their results.

Author response: We appreciate the care in this comment and line-item edits. We are excited that this is our first author's –first- manuscript. The senior author has re-edited for style and clarity.

With regard to the first result on mouse eye movements this new high frequency component is of the order of 6 microns on the retina and I feel appropriate and sophisticated measurements were done to quantify these eye movements. As far as I could determine from the relevant literature, the mouse retinal ganglion cell receptive fields are much larger than this. I believe an ocular movement has to be on the order of 1/2 receptive field in size to likely be of physiological import and these movements are much smaller than that. So they exist and are now newly reported but I personally doubt their physiological significance.

Author Response: The reviewer is correct regarding the approximate size of mouse RGC receptive fields. We would contend however, that neither the reviewer -nor we- can be certain of the physiological relevance of such eye motion on the limits (or benefits) to pattern vision (reference to work by M. Rucci et colleagues). The relevance of such findings will likely follow the observations we report here. In the spirit of scientific discovery, we describe this low magnitude motion for the first time and are hopeful that it spawns further investigation that asks the critical question the reviewer poses. If the magnitude of the eye motion is not impressive in its own right, we encourage the reviewer to examine the supplementary video file 5 and 10 to see visibly just how much this motion is suppressed after anesthesia. It would be empirically surprising if this motion did not impact vision in some way (either in a beneficial or consequential).

As a final point, we emphasize in the text that regardless of the consequence on mouse visual performance, the fast eye motion characterized here at over 150 Hz has the potential to blur any retinal imaging camera with exposure times over 6.67 milliseconds (corresponding to 30-150Hz eye motion). The image blur (regardless of optics, would blur such an image by 6-12 microns due to motion blur alone which is appreciable considering the target size of many neurons and neurites in the retina at or below 1 micron in diameter.

Thanks to the reviewer, we have included the following text in the Discussion that underscores this important point.

"Finally, it is worth noting that beyond the study of such eye motion in the pursuit of understanding neural function and control, accounting and correcting for eye motion is essential in achieving high resolution images of the retina. Left uncorrected, it will induce motion blur in high-resolution imaging modalities with exposure times more than ~5ms (200 Hz frame rate). And while flash photography could mitigate such blur, there are few video acquisition ophthalmoscopes to date that achieve these imaging speeds, especially for the mouse. This would mean that even through perfect optics, the awake retinal image would be blurred on the scale of 10-20 micrometers. Therefore, high-resolution imaging needs to achieve both aberration correction and high-speed imaging/registration >200Hz to achieve diffraction-limited potential in the awake mouse."

With regard to retinal blood flow being specifically reduced by anesthesia, I don't doubt that it is given the large decrease in pulse rate and blood pressure. I would ideally like to have addressed to what degree a 'purely passive' system would have had decreased blood flow given the reduction in the relevant cardiovascular parameters induced by anesthesia. Also I feel the description of the calculation of the blood flow was deficient given the great detail into which the paper went on how blood velocity and vascular diameter were measured. Again, all well done at the highest level of current retinal imaging technology.

Author Response:

The reviewer poses a relevant line of questioning. However, the number of parameters that must be measured and accounted for in such modeling is far beyond the scope of the current report. Heart rate, cardiac output, stroke volume, arterial O₂, CMR_{O2} and the differential shunting of the CNS relative to systemic flow must be accounted for in passive models of blood flow to be even close to approximation for the questions that the reviewer asks. Active regulation of this system which is known to exist in the eye complicates matters even further. We hope the reviewer agrees such questions are relevant for a future and detailed publication that seeks much different questions than our report here. Our objective is to show that single cell blood flow may be measured and consistent with previous reports in the brain, eye perfusion is impacted by ketamine/xylazine anesthesia in the retina.

We do agree however, that the description of the blood flow analysis was lacking detail. We therefore briefly describe the fundamentals of the measurement in the revised methods section. We also directly point the reader to our detailed 2019 eLife

paper that describes the calculation in detail (Joseph et al 2019, eLife). Finally, we have followed the suggestion above and have isolated blood flow data in a new more detailed figure (Fig. 9 in current report). Additional detailed description about the blood flow measurement was also added in the Methods section.

“Retinal blood flow suppressed by ketamine/xylazine anesthesia

Blood flow was measured by using the previously reported technique in the head-restrain mice³⁶. Briefly, when performing line-scanned imaging across a blood vessel, the flowing blood cell generates an angled trajectory in the space-time image indicating a change in position over time scaled by the angle of incidence of the beam across the vessel³⁶. The velocity of the blood cells can be quantified by measuring the slope of blood cell profiles (Fig. 9a and Fig. S6). Flow was found to be pulsatile (Fig 9b) but at different frequencies and velocities corresponding to variable heart rate with anesthesia (481 and 758 Hz in two awake mice), corresponding to the much higher cardiac frequencies³⁷ compared to that measured in anesthetized mice (271.2 ± 1.2 Hz, $n=5$ blood vessels from 3 mice). Blood flow in a single venule was reduced by 43% relative to baseline measure (1.56 $\mu\text{L}/\text{min}$ to 0.89 $\mu\text{L}/\text{min}$) 20 minutes after anesthesia (Fig 9c). Independent measures showed an abrupt narrowing of the vessel diameter right after the K/X injection, while the change of blood velocity was minor at the beginning but substantially drops over time after 20 mins K/X injection. These two parameters did not show equivalent changes which underscores the importance of direct measurements to reveal a complete picture of blood flow and its regulation³⁸. While eye motion (discussed above) gradually returns 80 minutes after K/X induction, blood flow remained low (0.66 $\mu\text{L}/\text{min}$) which was 58% below baseline. These findings suggest the return of eye motion preceded the return of retinal blood flow indicating return of physiological function occurs at different rates in the same mouse after K/X anesthesia. Flow measurements from the awake mice ($n = 5$ vessels from 3 mice) were also compared to previously reported data from our group in anesthetized mice ($n=123$ vessels from 20 mice, Fig. 9d)^{15, 19}. Despite single vessels where velocity was reduced under anesthesia, we find that overall, the flow vs vessel diameter relationship was generally similar. Fitted model diameter-flow curves (detailed in Methods) show minor overall differences in total blood flow.”

Additional methods section of blood flow measurement:

“Blood flow measurement

Single-cell blood flow was measured noninvasively in awake mice using an approach which has been described extensively in our previously publication¹⁵. Briefly, blood vessels were obliquely scanned in 1D (15 kHz) by freezing the slow (galvo) scanner and consecutive scans were concatenated through time, to form a space-time image (Figure S6 a). Blood cells appear as diagonal streaks, indicating their position across the scan. The slope of the streak combined with the angle between the direction of flow and the position of the 1D scan (extracted from subsequently acquired 2D raster image) were used in an automated algorithm employing the Radon transform to extract blood cell velocity. In our space-time images, streaks approaching a vertical orientation represent higher velocity

cells. The radon algorithm operates by splitting space-time images into overlapping ROI's. Each ROI is iteratively rotated 180° and a 1D intensity projection is extracted for each iteration. The standard deviation was calculated for each 1D projection and the highest value indicated the dominant angle and thus yielding a velocity for the ROI. The measurement bandwidth of this technique was 0.03-1275 mm/s, which was more than sufficient to measure biologically possible blood cell speeds in the awake-behaving mouse retina. Pulsatile flow was measured with high spatio-temporal resolution and overlaid as a velocity color map on the space-time image (Fig S6 b). To determine vessel lumen diameter, motion contrast images of the blood vessel were generated from 2D Cartesian images. The imaging of forward and multiply-scattered light through the RBC column within microvessels spanning <50 μm in size helped obtain an accurate measurement of the lumen diameter. Combined, the velocity and diameter measurements gave the mean flow rate (in μL/min) through each vessel. These measurements were done simultaneously with those of eye movements described in the section above. Flow-diameter model curves were fit to the measured data compare the two populations. Briefly, the model used was of the form $y=ax^b$. where y is flow rate (μL/min) and x is diameter (μm). “

With regard to mouse retinae thickening with anesthesia, there does seem to be an effect largely seen in mice beyond 80 minutes of anesthesia but the effect is not terribly large though it does seem to be statistically significant. So this may be important in sequential measurements of mouse retinal thickness if an experiment is done under anesthesia and is carried out for extended periods of time but there seems to be little effect of lesser durations of anesthesia.

Author Response: Many in vivo retinal imaging studies do not report when data is collected relative to anesthesia induction. In our own experience, many preparations require long-term physiological measurements lasting 1-2 hours. Dark adaptation, optoretinography, intrinsic signal optical imaging and blood flow challenges often try to maximize imaging time to perform repeat experimental trials or recover basic physiological responses. Moreover, other data are collected by users that are still gaining experience and may take extra time to perfect the experimental preparation (eg. Trainees that are perfecting their technique). Therefore, we feel this is an important observation that has not been reported and must be accounted for in a burgeoning field that seeks to perform a number of long-term measurements.

REVIEWERS' COMMENTS:

Reviewer #1 (Remarks to the Author):

The authors have correctly answered the questions raised. They have provided convincing illustrations and arguments. The changes made improve the article which can be published in Communications Biology.

Reviewer #2 (Remarks to the Author):

This is my second review of this paper. I made a number of general comments as well as many specific comments on the first version including comments about certain of the figures. I feel that both my general comments were well addressed by the authors and that both language and figure comments were taken to heart and thoroughly and appropriately dealt with. I therefore feel the paper is considerably improved and is now suitable for publication and I thank the authors for their high quality work in the advancement of this field.